# Impact of Six Months of Three Different Modalities of Exercise on Stress in Post-Treatment Breast Cancer Survivors

**DOI:** 10.3390/cancers16193398

**Published:** 2024-10-04

**Authors:** Daniel C. Hughes, Jessica Gorzelitz, Alexis Ortiz, Lorenzo Cohen, Dorothy Long Parma, Terri Boggess, Nydia Tijerina Darby, Shragvi Balaji, Amelie G. Ramirez

**Affiliations:** 1Institute for Health Promotion Research, University of Texas Health—San Antonio, 7411 John Smith Drive, Suite 1000, San Antonio, TX 78229, USA; longparma@uthscsa.edu; 2Department of Health and Human Physiology, University of Iowa, 115 S. Grand Ave., 110 IBIF, Iowa City, IA 52245, USA; jessica-gorzelitz@uiowa.edu; 3Physical Therapy Program, Allen College UnityPoint Health, 1825 Logan Avenue, Waterloo, IA 50703, USA; alexis.ortiz@allencollege.edu; 4Integrative Medicine Program, The University of Texas M.D. Anderson Cancer Center, 1515 Holcombe, Houston, TX 77030, USA; lcohen@mdanderson.org; 5Exercise and Sport Science Department, St. Mary’s University, One Camino Santa Maria, San Antonio, TX 78228, USA; tboggess@stmarytx.edu; 6Nydia’s Yoga Therapy & Open Hand Institute, 1403 Blue Crest Lane, San Antonio, TX 78232, USA; nydia@openhandinstitute.com; 7Baylor College of Medicine, 1 Baylor Plz, Houston, TX 77030, USA; shragvi.balaji@bcm.edu; 8Department of Population Health Sciences, Institute for Health Promotion Research, University of Texas Health—San Antonio, 7411 John Smith Drive, Suite 1000, San Antonio, TX 78229, USA; ramirezag@uthscsa.edu

**Keywords:** cortisol, stress, exercise, yoga, breast cancer

## Abstract

**Simple Summary:**

Research continues to show that exercising at recommended levels is associated with lower mortality rates and decreased risk of recurrence in breast cancer survivors (BCS). Moreover, those BCS exercising at levels meeting guidelines have higher self-reported health-related quality of life (HR-QOL) than those that do not. This includes lowered levels of perceived global stress. However, which modalities of exercise are most effective has yet to be determined. Our study investigates changes in stress experienced by breast cancer survivors participating in one of three different exercise modalities conducted over six months, as indicated by self-report and salivary cortisol measures. The revealed improvement in constructs of Health Related-Quality of Life (HR-QOL) suggest that engagement in exercise had a greater impact on outcomes than any particular modality.

**Abstract:**

Background/Objectives: Extensive evidence suggests that exercise is physically and mentally beneficial for cancer survivors. This study reports on changes in self-reported stress, physiological biomarkers for stress (salivary cortisol), and HR-QOL constructs for fifty breast cancer survivors participating in one of three different exercise programs over 6 months. Methods: Fifty post-treatment breast cancer survivors were randomized to either therapeutic yoga-based exercise (YE), comprehensive exercise (CE) (aerobic, resistance, flexibility), or choosing (C) their own exercise activities. Participants completed the Perceived Stress Scale (PSS), Medical Outcomes Short-Form 36^®^ (SF-36), and the Pittsburgh Sleep Quality Index (PSQI). Five samples of salivary cortisol were collected on two consecutive days. The 10 samples were used to calculate the diurnal rhythm slope. Outcome measures were repeated after six months. Results: All groups improved in HR-QOL measures of PSS; PSQI sleep quality components of latency and daytime functioning; and five of the ten SF-36 scales (Mental Component Scale, Social Functioning subscale, Mental Health subscale, Physical Component Scale, Physical Functioning subscale). Although the CE group observed the most favorable change in cortisol (−0.183), where cortisol slope changes approached significance (*p* = 0.057), but no significant decrease in cortisol between groups were noted. Conclusions: Our results suggest that it is the engagement of, rather than the specific type of exercise, which is associated with improved HR-QOL. However, longer-term studies with better adherence monitoring and larger sample sizes are needed to better determine clinical impact.

## 1. Introduction

Breast cancer remains the most prevalent cancer for women, with over 310,720 women expected to be diagnosed with breast cancer in 2024 [1]. Studies support a decreased risk of recurrence and lower mortality rates for those who are more physically active [2,3]. The stress of cancer can negatively impact health-related quality of life (HR-QOL), including physical and mental functioning [4,5,6]. Exercise for BCS is effective for improving HR-QOL [7,8]. However, the modality of exercise that is the most effective has yet to be determined.

Beyond daily stressors, BCS have specific cancer experiences, including, but not limited to, continued side effects of medical treatments [9] and often a fear of recurrence [10]. Not surprisingly, many BCS report more stress than those without a history of cancer [4,6]. Chronic stress has been associated with negative long-term health outcomes [11], including secondary cancer risk [12,13,14]. A perceived stressor stimulates the sympathetic adrenal-medullary (SAM) axis and activates the hypothalamic, pituitary, and adrenal (HPA) axis. Cortisol, an end product of the HPA axis and a physiological indicator of the state of the stress response, can be accurately measured in saliva [15,16]. Cortisol levels follow a circadian rhythm peaking upon awakening, then gradually decreasing throughout the day, and ultimately reaching their lowest levels at night [15,16]. Chronic stress can disrupt this rhythm [11]. Circadian rhythm reflects the capacity of the physiological system to adjust appropriately [17]. Cortisol rhythm is often dysregulated in cancer survivors [18]. Sephton and colleagues reported that dysregulated cortisol rhythm in BCS was associated with higher mortality [19].

Though exercise is effective in improving HR-QOL and perceived stress, less is known about which specific type of exercises have the best impact on the physiological effects of stress response [20]. This study examined changes in stress (self-reported and salivary cortisol circadian rhythm) and HR-QOL measures in post-treatment BCS participating in a six-month exercise program. Participants were randomized to either a yoga-based exercise program (YE); an individually tailored, comprehensive (aerobic+ resistance+ flexibility) exercise program (CE); or chose their own exercise (C). The purpose of this study was to determine which of the three exercise programs, YE, CE, or C, would be associated with the most improved stress and HR-QOL outcomes when compared to the other two.

## 2. Materials and Methods

### 2.1. Participants

Participants interested in the study called the phone number shown on the recruitment material, (ThriveWell^®^ Cancer Foundation; San Antonio, TX, USA). Of the 130 interested women, 121 met the inclusion criteria, and 94 participants provided informed consent and completed baseline fitness assessments, and 72 participants completed 6-month post-assessments. Of those 72, 50 provided adequate, (at least 3 specimens per day of collection), salivary cortisol samples for analyses, see Figure 1 This study is a secondary analysis of project IMPACT [21,22,23], with this study focusing specifically on HR-QOL stress-related measures and cortisol rhythm analyses.

### 2.2. Randomization

After baseline assessment, an adaptive minimization randomization technique was used to balance participant groups based on covariates of age, body mass index (BMI), and cardiorespiratory capacity. Participants were randomized to either a (1) Yoga-based Exercise program (YE) group); (2) individually tailored, Comprehensive Exercise program (CE) group; or, (3) a Comparison group (C), with their own choice of exercise. All groups were asked to complete three hours of exercise per week for the six-month active portion of the study.

### 2.3. Exercise Programs

#### 2.3.1. Yoga-Based Exercise Program

A structured Hatha Yoga exercise program was developed specifically for people with limitations of physical functionality, a characteristic of BCS. The protocol was developed by a certified yoga instructor who is also a licensed physical therapist. The program emphasized breath awareness (pranayama); a modified Sun Salutation; and postures (asanas): standing, seated, quadruped, twisting/rotation, prone, supine postures and transitions used between postures; modified inversion; and resting postures. The practice lasted approximately 50 min, including guided relaxation one body part at a time (savasana) in the final ten minutes. The protocol and sequencing of postures were designed so the participants could perform the same routine, regardless of the instructor or class. YE participants were asked to attend three 1 h yoga classes per week in a local yoga studio and were neither encouraged nor discouraged from other types of exercise. An audio CD and instruction booklet were provided for use at home when class attendance was not feasible.

#### 2.3.2. Comprehensive Exercise Program

For the CE group, tailored exercise programs were prescribed by the PI, an ACSM Clinical Exercise Physiologist^®^. Prescriptions were based on the participants’ individual fitness results and public health physical activity guidelines for adults [24]. The exercise programs included aerobic, resistance, and flexibility training for three one-hour sessions per week. The primary aerobic component was brisk walking; though, if preferred, cycling, treadmill walking, or jogging could be substituted. The initial recommendation was to complete at least 10 min continuous aerobic bouts, with the goal of accumulating at least 30 min per day of moderate to vigorous activity on five days per week. A packet of three different resistance bands was given to each CE participant as well. They were shown how to do a basic set of eight exercises using either their body weight or resistance bands for resistance. The sequence of exercises involved engaging major muscle groups first, then smaller muscles in a recommended ordered sequence of exercises for the stomach, back, torso, thighs, calves, shoulders, biceps, and triceps. The exercises were repeated for 8–12 controlled repetitions per set. Participants were asked to perform two sets on two days per week. For flexibility, participants were shown how to do complete body stretching, holding each stretch (static-non-ballistic) for at least 30 s. Participants were encouraged to log their exercises. (The logs were not collected.)

#### 2.3.3. Comparison Group

C group participants were asked to participate in exercise(s) of their choosing for three hours per week for six months. They were given the DIVA class schedule, which included over 30 exercise classes (aerobics, strength training, yoga, Tai Chi, Zumba, water aerobics, and belly dancing) held every week at different locations.

A member of the research team called each participant every two weeks to answer questions, monitor safety concerns, and encourage participation. A log of each conversation was kept. If any concerns were expressed, the staff member notified the PI was notified. The PI contacted the participant to follow-up on the concern, offer suggestions, and, if necessary, adjust the exercise program.

### 2.4. Measures

Prior to randomization, participants completed physical fitness/functioning tests and a psycho-social questionnaire packet that included: demographics; medical history; health-related quality of life measures (Medical Outcomes Study 36-Item Short-Form Health Survey [SF-36^®^] [25]; sleep quality (Pittsburg Sleep Quality Index [PSQI]) [26] and; perceived stress (Perceived Stress Scale; PSS) [27]. Participants were also asked to start collecting saliva samples the day after baseline assessment. Five samples of salivary cortisol were collected on two consecutive days. All outcome measures were repeated after six months.

#### 2.4.1. Fitness Assessment

The fitness/functional assessments included tests for cardiorespiratory capacity, muscular strength, flexibility, and body composition. Assessment procedures are summarized here; details have been reported previously [23]. Body composition was assessed by calculating body mass index (BMI) (kg/m^2^) and conducting a three-site (triceps, suprailium, and quadriceps) skinfold assessment.

For cardiorespiratory capacity, a ramped sub-maximal cycle ergometer protocol was used to estimate VO_2_max based on American College of Sports Medicine guidelines for submaximal exercise [28]. VO_2_max was estimated by linear regression analysis, regressing heart rate against corresponding VO_2_ uptake (mL O_2_/kg/min) levels during the period from the beginning of the exercise to the exercise phase ‘‘termination point’’ heart rate of 85% of the age-predicted maximum rate. The linear relationship was then extrapolated to the participants’ age-predicted maximum heart rate (220—current age), and the estimated VO_2_max level was then determined by the regression equation.

Arm, shoulder, and torso muscular strength were tested using a strength dynamometer. For leg strength, participants performed a timed sit-to-stand task. Hip and lower back flexibility were assessed using a sit-and-reach flexibility box. Upper body flexibility and balance were assessed using a forward-reach functional task. Participants received a $25 gift card upon completion of the fitness assessments.

Improvements in fitness on these assessments have been previously published [23]. Participants improved in all outcome measures in the expected direction with the exception of weight, cardiorespiratory capacity, and systolic blood, with essentially no change in these outcomes. Though weight remained essentially the same, significant improvements were seen in body composition with a reduction of % body fat, improved in sit-to-stand leg strength repetitions, and forward reach.

#### 2.4.2. Medical History

Medical information was used to calculate a co-morbidity index. Participants were asked if they had ever had any of the 17 health conditions listed: (diagnosis of a heart attack, heart failure, heart condition, circulation problems, blood clots, hypertension, stroke, lung problems, diabetes, kidney problems, rheumatoid arthritis, osteoarthritis, anemia, thyroid problems, neuropathy, fibromyalgia, and hepatitis). These individual responses were added to denote summed to calculate a co-morbidity index. This has been used with other female survivor groups [29] as an indicator of chronic conditions carried forward into the program. Participants were also asked to provide their cancer history, including time since diagnosis, type and stage of cancer, treatment (surgery, radiation and/or chemotherapy), time since last treatment, secondary cancers, and any post-treatment complications of lymphedema as best they could recollect. We did not repeat collecting medical history at six months at post-assessment.

#### 2.4.3. HR-QOL

##### Medical Outcomes Study 36-Item Short-Form Health Survey (SF-36^®^)

Participants completed a 36-item generic assessment for health concepts over the previous four weeks [25]. The 36 items are loaded on eight subscales: physical functioning (PF), role limitation due to physical health (RH), bodily pain (BP), general health (GH), vitality (VT), social functioning (SF), role limitation due to emotional problems (RE), and mental health (MH). The 36 items as well as the subscale items can also be aggregated into the Physical Component Scale (PCS) and Mental Component Scale (MCS). A typical question would be, “During the past 4 weeks, have you had any of the following: a. Problems with your work or other regular daily activity as a result of your physical health?” Responses are either: “YES” or “NO”. Scales for the SF-36^®^ have been normed for the general and specific US populations [25,30].

##### Pittsburg Sleep Quality Index (PSQI)

The PSQI assesses perceptions of sleep quality over the previous month. Nineteen items generate seven component scores: (1) subjective sleep quality; (2) sleep latency; (3) sleep duration; (4) habitual sleep efficiency; (5) sleep disturbances; (6) use of sleeping medication; and (7) daytime dysfunction. A typical question is: “During the past month, how would you rate your sleep quality overall?” Response choices are: “0 = very good”; “1 = fairly good”; “2 = fairly bad”; and “3 = very bad”. These yield a global score ranging from 0 to 21, with higher scores representing poorer sleep quality. The PSQI has been successfully tested and validated with cancer patient populations [31].

##### Perceived Stress Scale (PSS)

The PSS is a measure of global perceived stress to the degree that life circumstances are appraised as being unpredictable, uncontrollable, and overwhelmingly stressful over the previous 4 weeks. A typical question is “In the last 4 weeks, how often have you felt that you were unable to control the important things in your life?” Response choices are: “0 = never”; “1 = almost never”; “2 = sometimes”; “3 = fairly often”; and “4 = very often”. Higher scores represent higher perceived global stress and have been associated with increased susceptibility to illness and greater vulnerability to depressive symptoms in response to stressful life events [27]. The PSS has been normed for the general population using a probability sample of 2387 adults [32].

#### 2.4.4. Salivary Cortisol

Fifty participants (YE = 14; CE = 18; C = 18) provided saliva samples for both pre and post time points. Salivary cortisol has been confirmed to reflect free levels of circulating blood cortisol [15]. Whereas plasma levels reflect the level of bound and free cortisol, saliva samples assess levels of free cortisol or the more metabolically active levels. Specimens were collected using Salivette kits (Sarstedt, Inc. Newton, NC, USA) consisting of a cotton swab in a polypropylene vial. Participants provided five specimens per day for two consecutive days. The sample times were upon waking, 45 min later, 8 h after waking, 12 h after waking, and before going to bed.

Participants were given written instructions, briefed on how to collect the samples, and practiced collection of a practice sample under observation by a research team member before leaving the baseline assessment. Salivette kits were color-coded for sample times to match the instructions. Collected samples were mailed back, labeled, cataloged, and stored in a −80 °C freezer, and collectively shipped to Kirschbaum Laboratories in Dresden, Germany, for analysis. Concentrations were determined by immunoassay using an in-house DELFIA system (Pharmacia, Uppsala, Sweden) [33]. Results for each sample were reported in nmol/liter.

#### 2.4.5. Treatment of Data

All analyses were performed using Statistical Package for the Social Sciences (SPSS version 25.0, 2018; IBM Corp., Armonk, NY, USA). Descriptive statistics were performed for all continuous variables; percentages were calculated on categorical variables. Analyses were conducted for the participants, providing adequate samples at both time points (*n* = 50). Analyses examined the main effect of time collapsed across groups, and differences between groups (YE, CE, C) were considered exploratory analyses. A post hoc power analysis of 50 participants randomized to three groups, with PCS as the outcome variable, with power of 0.80 and alpha set at 0.05, yielded an effect size of 0.45.

In order to test the main effect of time, paired-sample *t*-tests were performed to compare pre- and post-values for SF-36, PSQI, PSS, and salivary cortisol. For between group effects, a time by group interaction model was employed to analyze for changes in SF-36, PSQI, PSS, and salivary cortisol. Bonferroni post hoc tests were applied when statistical significance (*p* < 0.05) one-sided was reached.

Cortisol values were log-transformed prior to analysis. The slope of the diurnal changes was used as the primary outcome measure for physiological stress activation. Slopes were calculated for days for which there were a minimum of three samples. The diurnal slope was determined by regressing cortisol values on the time of day each sample was collected. Both days were used for the regression to calculate the slope of the line (i.e., *β*) and referred to subsequently as ‘slope’. Steeper (more negative) slopes are indicative of a normal diurnal rhythm, i.e., an expected reduction in cortisol pattern throughout the day [15,16]; flatter (i.e., less negative) slopes indicate an abnormal dysrhythmic diurnal response [11]; Slope is considered a more informative pattern of measure of the functioning of the stress response than either the average of the total samples or of the area under the curve because slope utilizes all the information in the multivariate data [16].

## 3. Results

Baseline characteristics are shown in Table 1. Participants had moderate levels of co-morbidities (2.3), were overweight (BMI = 28.8 kg/m^2^) and had a low cardiorespiratory capacity (19.8 mL O_2_/kg/min). Approximately half of the sample had invasive cancer (46%) or ductal carcinoma in situ (DCIS, 49.2%). Most of the participants were diagnosed in the earlier stages: Stage I (30%) or II (32%); 30% reported HER2-related cancer subtype, while 65% did not know their subtype. We collected time since treatment on a Likert scale: the minimum time was six months (reported by three participants), and the maximum time was more than two years. Most women (60%) reported more than two years. Nineteen of the 50 were on hormonal replacement therapy at the time the study started. Ethnicity was one-third Hispanic (30%), and race was predominately white (75%). More than half of participants (60%) had at least a bachelor’s degree. Approximately half (46%) were fully employed, and 35% reported being either retired or a homemaker.

Participants self-reported lower physical and mental functioning, as scored by the SF-36 PCS and MCS aggregate scales (M = 42.0, M = 39.0, respectively) than population norms (PCS M = 49.2); MCS M = 53.8) [30]. These lower functioning scores are also evidenced in the SF-36 eight subscales (Table 2) [30]. The participants self-reported general stress levels, as scored by PSS (M = 33.6) higher than norms for women (M = 20.2) [32]. The mean salivary cortisol slope was −0.547, which is steeper (considered more favorable) compared to women with metastatic breast cancer (range −0.09 to −0.21) [19,34]. There were no reported injuries in any group related to the six-month exercise program. There were no differences in baseline values for those that completed the “post” assessment compared to those participants that did not complete the post assessment.

HR-QOL indicators (SF-36), sleep quality (PSQI), and perceived stress (PSS) improved for all three groups. Six of the ten SF-36 scales improved, including BP, GH, SF (*p* < 0.001); MH subscale (*p* = 0.001); as well as the PCS and MCS aggregate scales (*p* < 0.001). For the PSQI, (Table 3), the “sleep latency” and “daily function disturbance” subscales improved between baseline and post assessments (*p* = 0.026; *p* = 0.010, respectively); with the global PSQI score also improving (*p* = 0.041).

Although the change was in the expected direction, the cortisol slope did not reach statistical significance (*p* = 0.209). There were no significant differences in outcomes in-between groups. Cortisol slope change approached significance between groups, (*p* = 0.057), with the CE group resulting in a steeper (more favorable) change in slope (−0.183) when compared with the other two groups (YE = −0.029; C = +0.037).

(The data that support the findings of this study are available from the corresponding author upon reasonable request.)

## 4. Discussion

This study investigated the impact of six months of three different modalities of exercise on HR-QOL outcomes, including stress via self-report and cortisol diurnal rhythm, with three different modalities of exercise.

As expected, participants reported higher stress levels and lower HR-QOL at baseline, when compared with the same-aged general population without a history of cancer [25,32]. The high initial PSS score (M = 33.6) overall was much higher than reported for the general population (M = 19.6, SD = 7.49), which is indicative of such high-level stress [32]. Regardless of group, perceived stress reported by participants decreased, which is consistent with other cancer survivorship studies [35,36].

For self-reported HR-QOL, participants also improved in six of the ten SF-36^®^ subscales, PSS score, and two of the seven PSQI subscales. Cortisol diurnal rhythm changed in the expected direction but did not reach statistical significance. There were no differences between groups in self-reported HR-QOL improvements, suggesting that the actual engagement in exercise behaviors may be more important than the specific type of exercise. The benefits that were observed in this six-month program have also been reported in other exercise interventions, reinforcing the endorsement of the HR-QOL-enhancing benefits for cancer survivorship [7,20].

Though we have reported results of fitness outcomes previously [23], as our focus was on investigating which modality of exercise is more effective, it is important to relate the physical fitness outcomes to the focus of this study—stress and HR-QOL. As we concluded, our results suggest that engagement in activity may be more important than the specific modality of exercise. One common benefit for our cohort was a reduction in body fat. As we reported, all three groups had reductions in body fatness with no differences between groups [23]. Though without larger sample sizes and better monitoring of exercise adherence it is impossible to discern exact training effects, the loss in body fatness could be an important contributor to our outcomes.

This change in body composition is important as obesity and obesity-associated endocrine output have been associated with breast cancer recurrence [1,37,38,39]. Moreover, studies suggest that physical activities’ effect on biological markers associated with breast cancer risk may not only be the direct effect of activity but also the result of a favorable effect on managing/reducing obesity [20,40,41]. This could also be a factor in other outcomes such as cortisol diurnal pattern and extrapolated to self-report HR-QOL.

Psychosocial effects on cancer survivorship may be moderated and possibly mediated by disruption of circadian rhythm [18]. A blunted slope indicates less favorable outcomes for BCS [19]. Previous work from Bower et al. also reported a blunted stress response in fatigued BCS [42]. The study population’s baseline mean cortisol slope of −0.547 suggests a somewhat ‘healthier’ cortisol pattern. The clinical significance is indeterminable without longer-term studies. Studies investigating other factors, such as social support on stress in women with metastatic breast cancer, have shown that diurnal cortisol slope was not associated either with quantity or quality of social support. However, feelings of belonging, positive appraisal, and tangible support subscales were associated with mean cortisol levels [43]. Thus, further research to identify the factors with immediate and long-term influence on stress levels, not only in cancer survivors, but other populations experiencing the stressors of chronic disease, is highly warranted [44].

## 5. Study Limitations

The results should be interpreted with caution, as there are several limitations to be noted. Exercise compliance was not recorded, which is a major limitation of this study. However, participants were asked to self-record their activity using the International Physical Activity Questionnaire (IPAQ) [45]. We provided no training, and the results were highly variable and not valid; therefore, we did not use the measure. However, we did record attendance in the yoga group but not in the others, the reason being resource limitation. The only sense we had of participation was in how they performed in the post-assessments. Use of objective measures, such as accelerometers or fit watches, would have been a better route, though again there was a limit in resources. We acknowledge this is a major limitation and a caution in interpreting our results.

Also, we had a relatively small sample size (*n* = 50) that provided enough valid samples to be analyzed. For the collection of biological samples, participants’ actual collection methods may not have been as compliant with the procedures as instructed. Including the timing of the specimens to the requested times. Any “off-time” collection could bias diurnal slope calculations. However, prior to leaving the assessment site, participants were briefed on when/how to collect the samples and the importance of collecting the samples at the designated times, as well as given an instruction sheet with color-coding the Salivette tubes. Participants practiced collecting an actual sample under observation from one of the research team members.

Another limitation was that participants were not blinded to the groups that they were in. There was a lot of cross-communication among participants, and knowing they were not blinded may have biased some of the self-reported results to be more of what was expected as an answer, though such bias would be difficult to discern.

This study has several strengths. Studies that involve exercise and cancer survivorship do not include self-report coupled with a physiological marker. Moreover, the self-reported data are distributed across a variety of indicators (e.g., sleep quality, social functioning, perceived stress, etc.), facilitates a more comprehensive understanding of exercise effects as demonstrated by the self-reported data. Additionally, this study also explored how different modalities of exercise influence HR-QOL outcomes. This is an area of needed research as exercise modalities differ as to which physiological systems are utilized [20]. The study findings suggest engagement rather than modality of exercise to assume a more significant role in HR-QOL improvement, at least for the participants of this study. Further research is crucial in assessing the clinical implications of such a claim.

## 6. Clinical Implications

Though the improvements seen in PSS, sleep quality, and SF-36 aggregate and subscales reached statistical significance, what could be considered more important is the clinical relevance. For the sleep quality subscales and the PSS, it is difficult to interpret what can be considered clinical significance. However, for the SF-36 subscales, a clinically meaningful change can be considered five points in the final scoring [46]. Using these criteria, participants reported clinically significant improvements for the subscales of bodily pain (∆ = 6.6), general health (∆ = 10.2), mental component aggregate scale (∆ = 7.8), social functioning (∆ = 18.9) and mental health (∆ = 5.9).

## 7. Conclusions

Engagement in exercise, and not necessarily exercise modality, improved HR-QOL in BCS. Larger-scale studies that incorporate different exercise protocols for specific cancers are needed to better understand the mechanisms involved, optimize self-reported HR-QOL, and assess clinical implications.

## Figures and Tables

**Figure 1 cancers-16-03398-f001:**
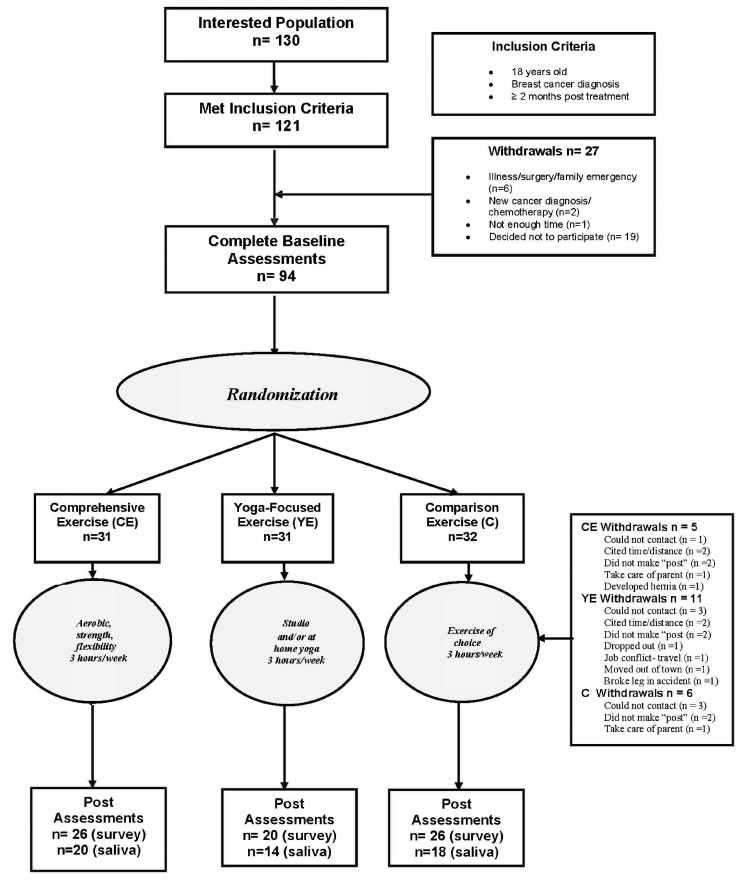
Study flow diagram.

**Table 1 cancers-16-03398-t001:** Participant baseline characteristics, mean standard deviation or *n* (%), *n* = 50.

Baseline Characteristic		All (*n* = 50)	YE (*n* = 14)	CE (*n* = 18)	C (*n* = 18)	*p*
Age		56.2 (7.9)	56.7 (9.6)	57.6 (6.6)	54.4 (7.0)	0.266
Co-Morbidity Index		2.3 (1.7)	2.1 (1.7)	2.1 (1.5)	2.7 (1.8)	0.323
Body Mass Index (BMI) (kg/m^2^)		28.8 (6.7)	29.1 (6.7)	29.1 (6.2)	28.1 (7.3)	0.810
VO_2_max (estimated) (mL O_2_/kg/min)		19.8 (5.1)	20.2 (5.6)	19.2 (4.9)	19.9 (5.0)	0.737
Lymphedema						
	Yes	19 (20%)	5 (16%)	6 (19%)	8 (25%)	
	No	72 (77%)	25 (81%)	25 (81%)	22 (69%)	
	Missing	3 (37%)	1 (3%)	1 (3%)	2 (6%)	
Ethnicity						
	Hispanic	30 (32%)	10 (32%)	7 (23%)	13 (41%)	
	Non-Hispanic	63 (67%)	21 (68%)	24 (77%)	18 (56%)	
	Missing	1 (1%)			1 (1%)	
Race						
	White	75 (79%)	26 (84%)	25 (81%)	24 (75%)	
	African American	5 (5%)	--	2 (6%)	3 (9%)	
	Asian	1 (5%)	1 (5%)	--	--	
	Other	11 (11%)	4 (13%)	3 (10%)	4 (12%)	
	Missing	2 (2%)	--	1 (3%)	1 (3%)	
Education						
	High school diploma	8 (8%)	3 (10%)	2 (6%)	3 (9%)	
	Technical	3 (3%)	3 (10%)	--	--	
	Some college	23 (25%)	7 (23%)	7 (23%)	9 (28%)	
	Bachelor’s degree	26 (28%)	10 (32%)	9 (29%)	7 (22%)	
	Master’s degree	28 (30%)	6 (19%)	10 (32%)	12 (38%)	
	Terminal degree (e.g., MD, PhD)	4 (4%)	2 (6%)	2 (6%)	--	
	Missing	2 (2%)	--	1 (3%)	1 (3%)	
Employment Status						
	Employed full time	46 (49%)	14 (45%)	15 (48%)	17 (53%)	
	Employed part time	8 (8%)	1 (3%)	5 (16%)	2 (6%)	
	Not working but seeking	2 (2%)	2 (6%)	--	--	
	Not working not seeking	1 (1%)	--	--	1 (3%)	
	Retired	22 (23%)	8 (26%)	8 (26%)	6 (19%)	
	Homemaker	13 (13%)	6 (19%)	3 (10%)	4 (13%)	
	Volunteer	1 (1%)	--	--	1 (3%)	
	Missing	1 (1%)	--	--	1 (3%)	

Note: Total % may not add up to 100% due to rounding. YE = Yoga-based exercise group; CE = Comprehensive exercise group; C = Comparison group; VO_2_max (mL O_2_/kg/min).

**Table 2 cancers-16-03398-t002:** Normative data, descriptive statistics and group differences for SF-36, PSS, cortisol slope, (*n* = 50).

Measure			Norms [25,32]	YE *n* = 14 Mean (SD)	CE *n* = 18 Mean (SD)	C *n* = 18 Mean (SD)	All *n* = 50 Mean (SD)	YE *n* = 14 Mean (SD)	CE *n* = 18 Mean (SD)	C *n* = 18 Mean (SD)	All *n* = 50 Mean (SD)	Differ *p*	Within Inter- Action *p*	Between Inter- Action *p*
SF-36														
	PCS		49.2 (15.1)	42.1 (3.6)	42.5 (3.4)	41.6 (3.5)	42.0 (3.4)	45.5 (4.5)	46.5 (6.3)	46.5 (5.1)	46.0 (5.3)	<0.001	0.688	0.847
		PF	50.7 (14.5)	50.7 (6.7)	51.8 (6.1)	49.8 (7.4)	50.8 (6.0)	51.2 (7.2)	51.3 (6.1)	52.2 (5.2)	51.6 (6.0)	0.335	0.321	0.947
		RP	49.5 (14.7)	25.5 (3.6)	25.8 (3.0)	24.6 (3.3)	25.3 (3.3)	27.1 (0.9)	26.0 (2.9)	25.7 (2.9)	26.2 (2.5)	0.102	0.570	0.305
		BP	50.7 (16.3)	38.8 (2.3)	39.1 (2.4)	37.7 (3.6)	38.5 (2.9)	42.4 (9.1)	44.2 (11.1)	48.0 (11.1)	45.1 (10.7)	0.307	0.162	0.531
		GH	50.1 (16.9)	43.8 (5.7)	42.9 (6.4)	45.1 (4.2)	44.0 (5.5)	55.2 (6.8)	55.2 (5.6)	52.3 (8.1)	54.2 (6.9)	<0.001	0.250	0.876
	MCS		53.8 (13.1)	39.8 (3.6)	38.8 (4.3)	38.8 (3.3)	39.0 (3.7)	50.4 (3.8)	45.8 (10.1)	45.5 (9.4)	46.8 (8.8)	<0.001	0.475	0.221
		VS	53.7 (15.4)	53.6 (4.2)	54.9 (6.0)	54.5 (4.5)	54.4 (4.9)	59.2 (7.8)	55.9 (8.8)	54.9 (11.5)	54.5 (9.6)	0.111	0.307	0.700
		SF	51.4 (13.9)	35.5 (3.5)	34.1 (2.8)	34.1 (3.4)	34.5 (3.2)	53.9 (7.3)	53.5 (7.5)	52.9 (6.7)	53.4 (2.0)	<0.001	0.927	0.758
		RE	51.4 (13.1)	19.0 (4.2)	18.5 (4.5)	17.9 (4.5)	18.4 (4.4)	19.8 (3.2)	19.0 (3.8)	19.8 (2.9)	19.5 (3.3)	0.106	0.595	0.844
		MH	54.3 (13.3)	53.7 (3.3)	53.9 (4.0)	53.0 (4.4)	53.5 (3.9)	63.0 (5.1)	57.8 (11.9)	58.3 (11.5)	59.4 (10.4)	0.001	0.429	0.337
PSS			20.2 (7.8)	32.7 (4.3)	33.6 (3.6)	34.3 (5.3)	33.6 (4.4)	14.4 (6.3)	16.5 (7.0)	20.1 (9.6)	17.2 (8.0)	<0.001	0.294	0.145
Cortisol Slope			N/A	−0.493 (0.22)	−0.573 (0.25)	−0.564 (0.20)	−0.547 (0.20)	−0.522 (0.20)	−0.756 (0.39)	−0.527 (0.21)	−0.608 (0.30)	0.209	0.121	0.057
			<Baseline (“Pre”)>					<6-Month (“Post”)>						

Note: YE = Yoga-based exercise group; CE = Comprehensive exercise group; C = Comparison group; PCS = SF-36 Physical Component Score; BP = SF-36 Bodily Pain subscale; MCS = SF-36 Mental Component Scale; MH = SF-36 Mental Health subscale; SF = SF-36 Social Functioning subscale; RE = SF-36 Role Emotional subscale; vs. = SF-36 Vitality subscale; PSS = Perceived Stress Scale; Cortisol slope (diurnal rhythm).

**Table 3 cancers-16-03398-t003:** Means, standard deviation and differences PSQI, (*n* = 50).

Measure		YE *n* = 14 Mean (SD)	CE *n* = 18 Mean (SD)	C *n* = 18 Mean (SD)	All *n* = 50 Mean (SD)	YE *n* = 14 Mean (SD)	CE *n* = 18 Mean (SD)	C *n* = 18 Mean (SD)	All *n* = 50 Mean (SD)	Differ *p*	Within Inter- Action *p*	Between Inter- Action *p*
PSQI												
	Global	5.30 (2.9)	5.94 (2.6)	6.88 (3.0)	6.14 (2.8)	4.10 (2.8)	5.45 (2.1)	6.38 (3.5)	5.48 (2.8)	0.041	0.686	0.189
	Total Disturb	7.69 (4.0)	9.22 (3.3)	10.3 (5.0)	9.22 (3.2)	8.23 (4.8)	10.2 (4.1)	10.8 (6.1)	9.91 (5.1)	0.283	0.931	0.225
	Mins-sleep	0.67 (0.6)	0.61 (0.7)	0.94 (0.8)	0.74 (0.7)	0.58 (0.7)	0.61 (0.5)	0.47 (0.7)	0.55 (0.6)	0.093	0.145	0.879
	Efficiency %	0.87 (0.1)	0.89 (0.1)	0.88 (0.1)	0.88 (0.1)	0.90 (0.1)	0.91 (0.1)	0.87 (0.1)	0.89 (0.1)	0.529	0.391	0.604
	Quality	0.71 (0.6)	0.78 (0.6)	0.94 (0.6)	0.82 (0.6)	0.57 (0.6)	0.89 (0.5)	0.83 (0.9)	0.78 (0.7)	0.637	0.523	0.469
	Latency	1.50 (0.8)	1.17 (0.9)	1.29 (0.8)	1.30 (0.9)	1.08 (0.9)	1.06 (0.5)	1.00 (1.0)	1.04 (0.8)	0.026	0.578	0.807
	Duration	0.46 (0.5)	1.00 (0.3)	0.89 (0.6)	0.81 (0.3)	0.62 (0.6)	0.78 (0.4)	0.94 (0.8)	0.79 (0.3)	0.971	0.183	0.074
	Efficiency	0.46 (0.7)	0.28 (0.6)	0.41 (0.9)	0.38 (0.7)	0.38 (0.5)	0.22 (0.6)	0.47 (0.7)	0.35 (0.6)	0.814	0.846	0.547
	Disturb	1.46 (0.5)	1.50 (0.5)	1.56 (0.6)	1.50 (0.5)	1.38 (0.5)	1.61 (0.5)	1.44 (0.5)	1.49 (0.5)	0.771	0.796	0.634
	Meds	0.43 (0.9)	0.28 (0.8)	0.89 (1.4)	0.54 (1.1)	0.29 (0.8)	0.44 (0.9)	0.89 (1.3)	0.56 (1.05)	0.954	0.668	0.168
	Function	0.57 (0.6)	0.94 (0.9)	0.94 (0.8)	0.84 (0.8)	0.29 (0.6)	0.61 (0.8)	0.67 (0.8)	0.54 (0.7)	0.010	0.974	0.236
		<Baseline (“Pre”)>				<6-Month (“Post”)>						

Note: PSQI = Pittsburg Sleep Quality Index; YE = Yoga-based exercise group; CE = Comprehensive exercise group; C = Comparison group; Global = Global score (sum of all components); Total Disturb = Total disturbance summary (Item #5); Mins-sleep = Minutes to fall asleep Item #2); Efficiency % = Sleep Efficiency % (hours sleep divided by hours in bed); Quality = Subjective Sleep Quality (Item #9); Latency = Sleep Latency; Duration = Sleep Duration; Efficiency = Sleep Efficiency; Disturb = Disturbance Score; Meds = Use of Sleep Aid Medications; Function = Daily Function Disturbance.

## Data Availability

The data that support the findings of this study are available from the corresponding author upon reasonable request.

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
