# Peer review of "Impact of Six Months of Three Different Modalities of Exercise on Stress in Post-Treatment Breast Cancer Survivors"

_cancers, 2024, doi:10.3390/cancers16193398_

Round 1
Reviewer 1 Report
Comments and Suggestions for Authors
In the present study by Hughes et al. (2024), the stress levels, salivary cortisol (a physiological stress marker), and health-related quality of life (HR-QOL) were examined in fifty breast cancer survivors who participated in various exercise programs over six months. The results showed that health-related quality of life improved in all groups regardless of the type of exercise, suggesting that engagement in exercise itself is crucial for enhancing health-related quality of life.
Overall, the study provides good insight into the role of physical activity on stress in breast cancer survivors. However, several points should be discussed to appropriately contextualize the study:
-
The major weakness of the study is that the exercise programs themselves were not thoroughly evaluated. There is no recording of training frequencies, etc. The authors mention this, but the study also lacks information on the training effects. The methodology describes a range of performance parameters that were collected, such as strength, endurance capacity, flexibility, and coordination. However, no results are reported. Was there any improvement? In this context, the absence of a waiting control group is also significant. While studies suggest that training can be effective, without precise control of training (training participation, training success, etc.) and without the presence of a control group, the results are very difficult to interpret. Statements like: “…regardless of the exercise type, suggesting that the engagement in exercise itself is crucial for enhancing…” are rather vague without a control condition. Therefore, the authors should clarify how the study's validity appears sufficient even without control of treatment.
-
In line 149, the IMPACT study is mentioned. Could you please cite this here with a source, if there are already publications?
-
Figure 1 shows the study flow diagram. What is the difference between "Lost to follow-up" and "Dropped out"? Also, please revise the figure as the arrows in the "Post" boxes are overlapping.
-
The methodology is overall well described. However, there are a few questions regarding Table 1. It mentions "VO2max." I am somewhat confused because section 2.4.1 refers to a submaximal exercise test. How can this determine VO2max? Please clarify this. What test was used to measure cardiorespiratory capacity? PWC test?
-
In section 2.4.2, various diagnoses are listed that were queried. These are not reported later (except lymphedema). It would be interesting to know which comorbidities still exist, as other diseases can also be related to clinical endpoints and can be influenced by physical activity.
-
Sections 2.4.6 and 2.4.8 have the same title and are content-wise related. Why is there a section in between? Can these not be combined?
-
Line 285: What are “adequate samples”? Please clarify this.
-
Line 290: Why was the main effect examined with paired t-tests and not determined using two-factor ANOVA, as was done for the interaction?
-
Table 1: What is the co-morbidity index? I could not find this in the methodology description. Please also ensure that units are included in the table.
Minor Comments:
- Line 75: HR-QOL was already abbreviated in line 70. Not necessary to abbreviate again.
- Line 81: “related” appears twice.
- Line 83: “Yoga” appears twice.
- Formatting of Table 2/3: In one case, “Pre” and “Post” are below and in another, above the table.
Reviewer 2 Report
Comments and Suggestions for Authors
The authors have undertaken a moderately interesting study. I think the acknowledgement of limitations is acceptable, however there are a few issues with the reporting.
1. I would be interested to see the follow-up physical test results with these perhaps included in the analyses.
2. How many women were on hormonal treatments at the time of the study?
3. What was the average (longest and shortest) time post treatment?
4. What was he exercise background of these women? Were they physically active before taking part?
This additional information would strengthen the manuscript.
Editing:
Page 2: Line 81 - 'constricts' should be 'constructs'
Page 2: Line 83 - what is 'yoga exercise yoga-based exercise'? Do you mean the latter?
Page 2: Line 92 - 'approached significance' would be better as 'approaching significance'
Page 2: Line 101: remove 'to be' before 'women'. It is not needed.
Reviewer 3 Report
Comments and Suggestions for Authors
Hughes and colleagues conducted a RCT to the impact of six months of three different modalities of exercise on stress in post-treatment breast cancer survivors. Overall the manuscript is well-written and this is an interesting study. I have a few comments that I think could help strengthen the presentation of the methods and results.
- 94 participants provided informed consent and completed baseline fitness assessments and 72 participants completed 6-month post-assessments, of those 72, 50 provided adequate salivary cortisol samples for analyses. This seems to be a large attrition rate that only half of the participants who consented remained in the analysis. Did this affect your study results? Participants who felt the program that was not helpful were more likely to drop off the study.
- in line 294, please indicate whether the pvalue used was two sided or one sided.
- in study limitation, please discuss that whether the fact that the patients were not blinded to their exercise affected how they report some of the health outcomes.
Author Response
Please see the attchment

Round 2
Reviewer 1 Report
Comments and Suggestions for Authors
Dear Authors,
Thank you for thoroughly addressing my previous comments. However, I would like to request a minor revision to further enhance the manuscript.
In lines 409-410, you mention that the post-assessment results provide indications of the training effect. This is an important aspect, yet it is briefly noted in the methodology section (lines 223-228). As this is a key limitation of the study and training is a central focus, I believe a more detailed discussion of this point is necessary.
I suggest expanding the discussion on how the positive effects differ depending on the type of intervention. What have other studies shown in this context? Were there previous recommendations regarding physical activity (strength training, endurance training, or daily activities) for this or similar populations? You mention that cardiorespiratory capacity did not improve, which is typically a primary focus of training programs. Could this suggest that general physical activity, rather than structured training, may be sufficient? Might the reduction in body fat be a crucial factor here?
These are just a few considerations that could be incorporated into the results discussion. Ultimately, you are evaluating training programs in your study and concluding that the type of activity may not matter. This point requires additional context and exploration. Currently, the discussion focuses on the improvement in target parameters but does not delve into the possible reasons behind these changes, which would also be valuable for practitioners in the field.
